# Arbuscular Mycorrhizal Fungi Enhance the Insecticidal Activity of *Annona muricata* L. Leaves

**DOI:** 10.3390/plants14223501

**Published:** 2025-11-17

**Authors:** Angela Michelle González-López, Evangelina Esmeralda Quiñones-Aguilar, Jhony Navat Enríquez-Vara, José Alejandro Martínez-Ibarra, Gabriel Rincón-Enríquez

**Affiliations:** 1Laboratorio de Fitopatología, Biotecnología Vegetal, Centro de Investigación y Asistencia en Tecnología y Diseño del Estado de Jalisco A.C. (CIATEJ), Camino Arenero 1227, El Bajío del Arenal, Zapopan C.P. 45019, Jalisco, Mexico; angonzalez_al@ciatej.edu.mx (A.M.G.-L.); equinones@ciatej.mx (E.E.Q.-A.); 2SECIHTI-Centro de Investigación y Asistencia en Tecnología y Diseño del Estado de Jalisco A.C. (CIATEJ), Camino Arenero 1227, El Bajío del Arenal, Zapopan C.P. 45019, Jalisco, Mexico; 3Laboratorio de Entomología Médica, Centro Universitario del Sur, Universidad de Guadalajara, Av. Enrique Arreola Silva No. 883, Ciudad Guzmán C.P. 49000, Jalisco, Mexico; aibarra@cusur.udg.mx

**Keywords:** arbuscular mycorrhizal fungi, annonacin, insecticidal activity, fall armyworm, triatomine bugs

## Abstract

*Annona muricata* (soursop) produces secondary metabolites with antimicrobial and insecticidal properties. Arbuscular mycorrhizal fungi (AMF) are known to enhance the production of secondary metabolites in medicinal plants. In this study, we aimed to evaluate the insecticidal activity of ethanolic leaf extracts from AMF-colonized soursop trees against the fall armyworm (*Spodoptera frugiperda*) and the triatomine bug *Triatoma pallidipennis*, a vector of Chagas disease. Ethanolic leaf extracts were obtained from trees inoculated with two AMF consortia (Cerro del Metate and Agua Dulce), with the species *Rhizophagus intraradices* and *Funneliformis mosseae*, and from non-mycorrhizal plants (SM). Extracts were tested in bioassays specific to each insect, including chemical and negative controls, and survival was analyzed using Kaplan–Meier curves. Extracts from plants colonized by *F. mosseae* exhibited insecticidal activity against *S. frugiperda,* causing 72% larval mortality, comparable to that of the commercial insecticide. In contrast, extracts from plants inoculated with the Agua Dulce consortium caused 65% mortality in *T. pallidipennis* adults. These extracts showed significantly higher annonacin content (µg·g^−1^ DW). Overall, the results demonstrate that AMF colonization can enhance the synthesis of metabolites such as annonacins and contribute to increased insecticidal activity in *A. muricata*. Our findings suggest AMF-assisted cultivation has the potential to enhance botanical insecticides.

## 1. Introduction

Soursop (*Annona muricata* L.), a member of the Annonaceae family, is cultivated in Latin America primarily for its fruit [1]. While its fruits are usually consumed fresh, different parts of the plant are also valued in traditional medicine for their therapeutic properties, particularly in the treatment of cancer and parasitic infections [2,3]. The antitumor activity of *A. muricata* extracts has been reported in numerous studies [4,5,6]. In addition, such extracts have shown toxic activity against some medically relevant protozoa such as *Leishmania braziliensis* and *L. panamensis*. Based on reported findings [7], ethyl acetate is more effective than the conventional antileishmanial drug Glucantime^®^. The results of other in vitro studies have demonstrated that stem and root extracts, in addition to their fractions, inhibit the growth of *Plasmodium falciparum*, the causative agent of malaria [8].

In recent metabolomic analyses, a wide variety of metabolites in *A. muricata* have been identified, with acetogenins and alkaloids being the predominant groups [3,9]. Among them, acetogenins such as annonacin and squamocin are the most biologically active compounds and are mainly responsible for the cytotoxic, neurotoxic, bactericidal, and insecticidal properties of the plant [3,10]. For instance, acetogenins obtained from *Annona cherimola* and *Annona montana* exhibited toxic effects against the milkweed bug *Oncopeltus fasciatus* [11].

Different parts of *A. muricata* plants have been used as insecticides and repellents [2]. In agricultural pests such as *Plutella xylostella*, *Anastrepha ludens*, *Bactericera cockerelli*, and *Sitophilus zeamais*, the larvicidal effects of *A. muricata* plant extracts have been reported [12,13,14,15]. In comparison, in insects of medical significance, such as mosquitoes (*Culex quinquefasciatus* and *Aedes aegypti*), extracts have shown both repellent and insecticidal effects in larvae and adults [16,17]. These findings highlight the potential of soursop extracts as botanical insecticides against a broad spectrum of pest insects.

AMF are known to enhance the production of secondary metabolites and influence the biosynthesis and accumulation of active ingredients such as terpenes, phenols, and alkaloids in plants [18]. AMF form symbiotic relationships with more than 70% of plant species, improving nutrient uptake and increasing tolerance to abiotic and biotic stress through the accumulation of secondary metabolites [19,20,21,22,23]. It has been documented that AMF naturally colonize soursop plants and influence growth and tolerance to abiotic stress [24,25,26]. However, despite the extensive evidence of AMF-mediated enhancement of bioactive compounds, little is known about how these symbiotic associations affect the synthesis of insecticidal metabolites. The interaction between AMF colonization, the regulation of secondary metabolite pathways, and the resulting insecticidal potential of plants remains scarcely explored. The authors of previous studies have mainly focused on the effect of AMF on medicinal or pharmacologically active molecules, with no studies addressing their influence on compounds with insecticidal properties.

Against this background, we hypothesize that association with arbuscular mycorrhizal fungi enhances the biosynthesis of acetogenins, thereby increasing the insecticidal potential of *A. muricata* leaves. Therefore, the aim of the present study was to evaluate the effect of AMF on the insecticidal activity of *A. muricata* leaves against two insects of agricultural and medical importance: the fall armyworm *Spodoptera frugiperda* and the kissing bug *T. pallidipennis* [27].

## 2. Results

### 2.1. Biological Activity of Ethanolic Extracts of Soursop Against Triatoma pallidipennis

The evaluation of the different treatments applied to *T. pallidipennis* adults revealed that cypermethrin (insecticide) induced the highest mortality, reaching 85% (χ^2^ test, *p* < 0.001; Table 1); in comparison, the untreated control exhibited only 15% mortality after 30 days. Mortality in the treated groups became evident from the second week onward. Among the plant extracts tested, significant differences in mortality were observed for *A. muricata* inoculated with the AD (65%) and RI (65%) consortia at 800 mg·mL^−1^ (*p* = 0.0012 for both); in comparison, the remaining treatments showed no significant differences relative to the control (*p* > 0.05). No significant variation in egg production per surviving female was observed among treatments (Kruskal–Wallis, *p* > 0.05, Table 1).

Survival curve analysis results revealed significant differences between the cypermethrin treatment and the untreated control (Log-Rank, *p* ≤ 0.001; Figure 1). A significant reduction in survival probability was also detected in the AD treatment GROUP at 800 mg·mL^−1^ compared with the control (Log-Rank, *p* ≤ 0.001; Figure 1). These results indicate that the most effective treatments against *T. pallidipennis* were AD at 800 mg·mL^−1^ and cypermethrin at 100 ng·insect^−1^.

No apparent sex-related differences were observed in the insecticidal response to soursop extracts and cypermethrin; therefore, data from both sexes were pooled for analysis. Although the annonacin concentration in the AD treatment group was not significantly different from the control, a numerical increase was observed, which may partly account for the insecticidal effect detected in *T. pallidipennis* adults (Table 1).

### 2.2. Biological Activity of Ethanolic Extracts of Soursop Against Spodoptera frugiperda

Survival curve analysis of *S. frugiperda* larvae exposed to soursop extract incorporated into the diet at 0.25 mg·g^−1^ revealed that both the FM treatment and Palgus 0.066 µg·g^−1^ (positive control) significantly reduced larval survival relative to the negative control (1% Tween 80; Log-Rank, *p* ≤ 0.05; Table 2; Figure 2). Both Palgus 0.066 µg·g^−1^ and FM treatment significantly reduced larval survival compared with the negative control (χ^2^, *p* < 0.001), resulting in 100% and 72% mortality, respectively. In contrast, the remaining treatments (Palgus 0.0066, CM, RI, AD, and SM) showed no significant differences relative to the control (*p* > 0.05).

After six days of feeding, larvae treated with Palgus 0.066 µg·g^−1^ and FM showed significantly higher weight compared with the negative control (Dunn, *p* ≤ 0.05; Table 2). At the pupal stage, individuals treated with the lower Palgus dose (0.0066 µg·g^−1^) exhibited a significant reduction in biomass relative to all other treatments (Tukey’s test, *p* ≤ 0.05; Table 2). In contrast, no significant differences in pupal biomass were observed among larvae treated with *A. muricata* extracts (*p* > 0.05).

Regarding developmental time, negative control larvae reached pupation in 19.0 ± 0.46 days; in comparison, larvae exposed to Palgus 0.066 µg·g^−1^ and FM developed faster (17.5 ± 0.55 and 15.8 ± 0.47 days, respectively; Dunn, *p* ≤ 0.05). In contrast, larvae treated with AD and RI extracts exhibited longer developmental times (20.1 ± 0.40 and 20.0 ± 0.39 days, respectively; Dunn, *p* ≤ 0.05). Significant differences among treatments were confirmed by means of the Kruskal–Wallis test (H = 48.23, *p* < 0.0001). For pupal duration, significant differences among treatments were observed (H = 19.86, *p* = 0.0029). Only the RI treatment differed significantly from the others, showing a shorter pupal stage (10.1 ± 0.24 days; Dunn, *p* ≤ 0.05) (Table 2).

### 2.3. Annonacin Quantification via HPLC (González-López et al., 2025 [23])

Annonacin was detected in all methanolic extracts from *A. muricata* leaves (Table 1). The highest concentration was observed in RI treatment (1209.0 ± 272.7 µg·g^−1^ DW), which differed significantly from the non-mycorrhizal control (SM) that exhibited the lowest content (705.1 ± 47.5 µg·g^−1^ DW; LSD test, *p* ≤ 0.05). These data, reported previously in [23], indicate that AMF symbiosis enhanced the biosynthesis or accumulation of annonacin in *A. muricata* leaves. Notably, treatments with higher annonacin levels (RI and AD) also corresponded to higher mortality in *T. pallidipennis* (Table 1)*,* suggesting a potential link between annonacin accumulation and insecticidal activity. Although this trend was not statistically significant in *S. frugiperda*, the RI and AD extracts still induced mortality levels close to 40% (Table 2). Interestingly, the *F. mosseae* (FM) extract, which contained an intermediate annonacin concentration (997.6 ± 130.8 µg·g^−1^ DW), produced the strongest larvicidal effect in this model.

## 3. Discussion

### 3.1. Biological Activity of Ethanolic Extracts of Soursop Against Triatoma pallidipennis

Comparable findings have been reported with *Azadirachta indica* extracts applied to *T. pallidipennis* nymphs, which exhibited insecticidal effects between the first and third weeks post-application [28]. Similarly, ethanolic extracts of *A. muricata* caused high mortality in *Rhodnius prolixus* (90%) and *R. pallescens* (76%) [29]; in comparison, *A. coriacea* and *A. reticulata* induced 33% and 35% mortality in *R. neglectus*, respectively [30].

Numerous studies have demonstrated that species of the Annonaceae family exhibit insecticidal activity against a wide range of arthropods [31,32,33], largely attributed to acetogenins, the main bioactive metabolites in *A. triloba*, *A. muricata*, and *A. squamosa* [10]. These species contain complex mixtures of up to 30 acetogenins, combined with alkaloids, sesquiterpenes, and monoterpenes [34]. Among these compounds, acetogenins have received the greatest attention. Álvarez et al. [11] demonstrated in their study that squamocin, annonacin, molvizarin, itrabin, and almunequin were the most potent molecules, exhibiting strong insecticidal effects against the milkweed bug *Oncopeltus fasciatus*.

Annonacin has been reported to exhibit insecticidal activity against different insect species [10], which may explain the effect observed in *T. pallidipennis* adults. In the present study, the highest mortality rates (65–72%) were associated with annonacin concentrations ranging from approximately 997 to 1209 µg·g^−1^ DW (treatments FM and RI, respectively). Annonacin acts as a potent inhibitor of mitochondrial complex I (NADH dehydrogenase), disrupting the electron transport chain and ATP synthesis in insect tissues [35]. However, not all acetogenins display equal potency, as structural variations (particularly in the number and position of tetrahydrofuran (THF) rings) strongly influence their insecticidal activity [35]. Given that the estimated topical doses in our assays (≈8–19 µg per insect^−1^) are comparable to those previously shown to induce physiological toxicity, it is plausible that AMF-mediated enhancement of annonacin contributed to the mortality pattern recorded in *T. pallidipennis*. Therefore, the higher mortality observed in *T. pallidipennis* may result from the combined effect of AMF-induced metabolic shifts and the presence of multiple bioactive acetogenins with differing activities. Nevertheless, it remains necessary to determine (1) whether annonacin is indeed responsible for the insecticidal effect observed, and (2) which other compounds, apart from annonacin, were enhanced by mycorrhizal inoculation and may also contribute to the insecticidal activity of the evaluated extract.

AMF symbiosis enhances phosphorous (*p*) and nitrogen (N) uptake [22]. *p* directly participates in the mevalonate (MVA) and methylerythritol phosphate (MEP) pathways, providing key precursors such as acetyl-CoA, ATP, NADPH, glyceraldehyde-3-phosphate, and pyruvate for essential routes for terpenoid biosynthesis. It also increases levels of high-energy phosphate intermediates such as isopentenyl pyrophosphate (IPP) and dimethylallyl pyrophosphate (DMAPP), essential for the biosynthesis of terpenoids and other active molecules [22]. In addition, enhanced N uptake through AMF improves carbon availability and regulates phenolic and anthocyanin biosynthesis by modulating transcription factors [36].

### 3.2. Biological Activity of Ethanolic Extracts of Soursop Against Spodoptera frugiperda

The authors of previous studies reported that methanolic extracts of *Annona coriacea* reduced the larval viability of *S. frugiperda* by 24% [37], which is consistent with the larvicidal activity observed in the present study. In the Annonaceae family, insecticidal and larvicidal activity has been primarily attributed to acetogenins [10]. For example, Di Toto-Blessing et al. [38] demonstrated that cis-annonacin-10-one and gigantetronenin induced larval mortality and reduced feeding in *S. frugiperda*; in comparison, annonacin produced 50% mortality without reducing feeding. Similar findings were reported by Guadaño et al. [39] in *Spodoptera littoralis* and *Myzus persicae*. In addition, Álvarez et al. [11] showed that squamocin caused 100% mortality in early instars; in comparison, other acetogenins (neoannonin, itrabin, almunequin, asimicin, motrilin, chemorimolin-2, and tucumanin) induced mortality rates of 10–30% in *S. frugiperda*. The larvicidal effect of FM-inoculated soursop may therefore be linked to the presence of acetogenins in *A. muricata* extracts, although only annonacin was detected in this study (Table 1). Since annonacin levels in FM-inoculated plants were statistically similar to the control (Table 1), it remains possible that other acetogenins contributed to the observed effect.

The higher activity observed in FM-inoculated plants may result from specific plant–AMF interactions. Although direct evidence of species-specific regulation of secondary metabolism by AMF is limited, some researchers suggest that different AMF species can differentially influence metabolite profiles in their host plants [40,41,42,43]. Similar AMF-specific metabolic modulation has been reported by Liu et al. [44], whereby *Rhizophagus irregularis* increased tannin content, phenol oxidase activity, and monoterpenes in *Perilla frutescens*, whereas *F. mosseae* reduced sesquiterpenes and showed neutral or opposite effects on the growth of *Spodoptera exigua*. In addition, *F. mosseae* is among the most frequently reported AMF species associated with enhanced production of secondary metabolites, likely through nutrient-mediated and signaling mechanisms involving jasmonate and phenylpropanoid pathways [45,46,47]. The increased larvicidal activity observed in FM-treated plants could therefore reflect a combined effect of these regulatory mechanisms and a species-specific phenomenon. Nevertheless, further metabolomic and signaling studies are needed to confirm the mechanisms hypothesized.

Interestingly, both Palgus 0.066 and FM treatments increased larval biomass after six days of feeding. This apparent increase should not be interpreted as an indicator of improved larval health, but rather as a potential compensatory response to dietary stress. Sublethal concentrations of insecticidal compounds are known to induce hormetic effects, stimulating feeding or growth as larvae attempt to offset toxicity [48,49,50]. Such effects are often species- and dose-dependent. This result contrasts with those of Alves et al. [51], who reported decreased larval weight with *A. coriacea* leaf extracts. For instance, sublethal methiocarb increased *S. frugiperda* larval weight after 12 days of feeding [52]; in comparison, Tasei et al. [53] observed that deltamethrin at sublethal doses stimulated feeding in *Bombus terrestris*. Based on the above findings, the increase in larval biomass observed in the FM treatment may represent a hormetic or compensatory feeding response to dietary toxicity; further confirmation is needed, however.

A similar trend was reported by Liu et al. [44], who found in their study that inoculation of *Perilla frutescens* with *F. mosseae* increased the size of *Spodoptera exigua* larvae compared with *Rhizophagus irregularis*, even in the absence of mortality. In their study, larvae fed directly on mycorrhizal leaves, suggesting that AMF-induced metabolic changes can modulate insect growth through altered secondary metabolite profiles. Accordingly, the increased larval biomass observed in the FM treatment in the present study may reflect metabolic adjustments associated with this symbiosis rather than improved larval health.

At the pupal stage, biomass was unaffected by *A. muricata* extracts, in contrast with the synthetic insecticide (Palgus 0.0066 µg·g^−1^), which significantly reduced pupal weight. This result contrasts with the findings of Freitas et al. [37], who reported reductions in pupal biomass with *A. coriacea* extracts. Regarding development, our results align with those of Freitas et al. [37], who observed differences in larval biomass and viability but not in pupal duration. Conversely, Álvarez et al. [11] reported prolonged larval development caused by ingestion of specific acetogenins (chermolin-2, itrabin, asimicin, motrilin, and almunequin). Thus, the longer larval development times observed with AD and RI treatments may be related to the presence of acetogenins, a hypothesis that requires phytochemical confirmation in future studies.

Similar results have been reported in other AMF–plant interactions. In *Mimosa tenuiflora*, inoculation with *Gigaspora albida* increased total phenolic content and enhanced the larvicidal activity of leaf extracts in *Aedes aegypti*; in comparison, plants inoculated with *Claroideoglomus etunicatum* showed no larvicidal activity [40]. In addition, in *Perilla frutescens*, colonization by *Rhizophagus irregularis* increased the levels of phenolic, tannin, and volatile compounds, resulting in reduced feeding by *Spodoptera exigua* [44]. Similarly, in *Trifolium pratense*, inoculation with *Claroideoglomus claroideum* and mixed AMF consortia elevated monoterpene levels, such as (S)-limonene and β-pinene, metabolites associated with pest deterrence [54].

This study provides evidence that AMF inoculation can influence the insecticidal activity of *A. muricata* extracts; however, the absence of full metabolomic profiling limits the identification of additional compounds. Annonacin was the only acetogenin detected; however, other metabolites may also contribute to the observed effects. Further metabolomics studies are needed to confirm these findings.

The combination of AMF mycorrhization and *Annona* extracts is a promising biotechnology strategy for pest management. Enhanced secondary metabolism by AMF can increase the efficacy of botanical insecticides. However, several limitations remain for large-scale applications. Annonacins have shown neurotoxic effects at high doses in animals [55], raising important biosafety concerns. Moreover, the extraction and standardization of *Annona* metabolites at scale are challenging due to their chemical instability and variability among plant sources. Field-level studies are necessary to evaluate their stability, persistence, and effectiveness under natural conditions. Therefore, AMF–*Annona* systems should be regarded as a complementary approach to integrated pest management.

## 4. Materials and Methods

### 4.1. Plant Material and Mycorrhizal Inoculation

Seeds of *Annona muricata* were obtained from fresh fruits collected in the main soursop-producing region of Compostela, Nayarit, Mexico, and were used to produce 60-day-old seedlings following the procedure described in [26]. Seedlings were transplanted into 15 L pots filled with 13.6 kg of sterile substrate consisting of a mixture of sand, soil, and perlite in a 6:3:1 ratio (v:v:v). The substrate was sterilized in an autoclave (121 °C, 1.0546 kg·cm^−2^, 6 h). At the time of transplanting, 100 spores of the consortia Cerro del Metate (CM) and Agua Dulce (AD), or of the species *Funneliformis mosseae* (FM) and *Rhizophagus intraradices* (RI), were directly applied to the roots.

Spores of *F. mosseae* were obtained from monosporic propagation in trap pots containing *Sorghum bicolor*, *Tagetes erecta*, and alfalfa (*Medicago sativa*). This isolate originated from the Cerro del Metate site in Tzitzio, Michoacán, and is deposited at the National Genetic Resources Center (Centro Nacional de Recursos Genéticos, CNRG-INIFAP; Tepatitlán, Jalisco, Mexico) under the strain name QR01 and accession code CM-CNRG-TB233.

Spores of *Rhizophagus intraradices* were obtained from propagation pots containing *Sorghum bicolor*, *Tagetes erecta*, and *Medicago sativa*, using a commercial inoculum known as Micorriza INIFAP [56].

The inoculum level (100 spores per seedling) was chosen based on prior experiments by our research group, wherein this dose ensured rapid and consistent AMF colonization in other plant species. In addition, 10 g of sterile sand was added to seedlings without AMF (SM). The consortia contain several AMF species, previously characterized in detail [57]. The CM consortium includes 13 AMF species, *Acaulospora excavata*, *Acaulospora rehmii*, *Acaulospora scrobiculata*, *Acaulospora spinosa*, *Acaulospora* sp., *Diversispora aurantia*, *Funneliformis geosporum*, *Funneliformis mosseae*, *Glomus deserticola*, *Glomus glomerulatum*, *Glomus microaggregatum*, *Septoglomus viscosum*, and *Dentiscutata erythropus*, with *G. glomerulatum* being the most abundant. The AD consortium contains nine AMF species: *Acaulospora schenckii*, *Acaulospora excavata*, *Acaulospora scrobiculata*, *Acaulospora spinosa*, *Acaulospora* sp., *Claroideoglomus claroideum*, *Funneliformis geosporum*, *Glomus deserticola*, and *Rhizophagus clarus* [57]. The AMF consortia CM and AD were originally obtained from the rhizosphere of *Agave cupreata* plants growing in natural and cultivated sites in the state of Michoacán, Mexico. Both consortia and AMF species are preserved in the Phytopathology Laboratory of the Plant Biotechnology Division at CIATEJ. The experiment consisted of five treatments (two AMF consortia, two AMF species, and a non-mycorrhizal control) arranged in a completely randomized design with seven replicates per treatment, with each replicate corresponding to one pot containing a single soursop plant. Inoculated plants were maintained under greenhouse conditions (27.8 ± 0.26 °C morning; 34 ± 0.15 °C afternoon; 56–41 % relative humidity; 56.5–22.2 µmol m^−2^ s^−1^ PAR) for 19 months and watered weekly with tap water.

After 19 months, root samples were collected to verify AMF colonization. Fine roots were cleared and stained following the method of Phillips and Hayman [58], with some modifications. Root segments (1 cm in length) were placed on a glass slide and examined under a compound microscope. The percentage of root colonization was determined following the method described by McGonigle et al. [59]. Colonization data correspond to values previously reported in [23], obtained under identical inoculation and growth conditions.

### 4.2. Preparation of Soursop Leaf Extracts

Leaves from *A. muricata* plants inoculated with AMF and from non-mycorrhizal plants (SM) were collected after 19 months. The leaves were frozen at −80 °C for 24 h and subsequently lyophilized in a Shin BioBase freeze dryer (Model TFD5503, Dongducheon, Republic of Korea) for 76 h. The lyophilized material was ground using an electric mill (Hamilton Beach, Glen Allen, VA, USA). For each treatment, 5 g of powdered plant material was placed in 50 mL Falcon^®^ tubes and extracted with ethanol to a final volume of 45 mL. The mixture of plant material and solvent was sonicated (Branson^®^ 2510; 40 kHz; 28–32 °C, Danbury, CT, USA) for 1 h and centrifuged for 30 min at 13,000 rpm. The supernatant was collected and stored for solvent evaporation, and the pellet was re-extracted with ethanol under the same conditions. This process was repeated three times. Supernatants from the successive extractions were pooled and evaporated in a Vacufuge Plus Eppendorf^®^ (Enfield, CT, USA) concentrator at 45 °C until complete removal of the solvent. Extraction yield averaged 25.7 ± 3.4 % (*w*/*w*) based on dry leaf weight. The resulting crude extract was stored at 4 °C and protected from light until use in the bioassays. These ethanolic extracts were used in all bioassays; a separate methanolic extraction was performed exclusively for HPLC quantification (see Section 4.4.2).

### 4.3. Bioassay in Triatoma pallidipennis Adults

#### 4.3.1. Insects

Adults of *T. pallidipennis* were obtained from a colony maintained at the Medical Entomology Laboratory of the Centro Universitario del Sur, Universidad de Guadalajara. Prior to use in the bioassays, the insects were fed on New Zealand White (NZW) rabbits using the technique described by Ryckman [60]. All procedures involving *T. pallidipennis* and NZW rabbits were conducted in accordance with the Mexican Official Standard NOM-062-ZOO-1999 (Guidelines for the Care and Use of Laboratory Animals) [61]. These procedures were authorized by the Ethics Committee for Research (Comité de Ética en Investigación), Coordination of Research and Postgraduate Studies, Centro Universitario del Sur, University of Guadalajara (approval no. CEI/119/2025).

#### 4.3.2. Application of Extracts

The application of ethanolic leaf extracts of *A. muricata* was carried out on *T. pallidipennis* adults following a methodology described previously [30], with slight modifications. Twenty triatomine adults were used per treatment (10 males and 10 females). Adult insects were randomly assigned to treatments using a randomization procedure to minimize selection bias. Extract concentrations of 400 and 800 mg·mL^−1^ were prepared in 100% DMSO from ethanolic leaf extracts of soursop plants inoculated with AMF consortia (CM and AD), AMF species (FM and RI), and non-mycorrhizal plants (SM). These concentrations were selected by considering the extract yield and practical feasibility for topical application. A negative control consisting of 100% DMSO solution was included, in addition to a positive control with cypermethrin (TUCAGRO^®^, Cipermetrina 200 CETM 200 g a.i.·L^−1^, Ecatepec de Morelos, Estado de Mexico, Mexico) dissolved in DMSO to achieve a concentration of 100 ng per insect.

For each treatment, 5 µL of extract, positive control, or negative control was applied to the pronotum of adult bugs using a micropipette. The insects were then placed individually in 250 mL natural polyethylene jars, each containing a filter paper disk lining the base. All insects were incubated at 26 °C and 60–70% relative humidity for 30 days. Mortality was recorded every three days.

Each treatment was applied to groups of five males and five females and the experiment was conducted twice. At the end of the assay, the number of eggs laid by each treated female was recorded.

### 4.4. Bioassay in Spodoptera frugiperda Larvae

#### 4.4.1. Rearing of *Spodoptera frugiperda*

A colony of *S. frugiperda* was established in the Insectary of the Entomology Laboratory at CIATEJ, Zapopan Unit, Jalisco, Mexico, from larvae collected in nearby maize fields. Insects were maintained and fed on an artificial diet following a procedure described previously [62] in a growth chamber at 28 °C, 65% relative humidity, and a 12:12 h (light–dark) photoperiod. To obtain second-instar larvae for the experiments, 15 females and 15 males were placed in paper bags and fed with a 15% honey solution soaked in cotton. Egg masses deposited on the paper bags were collected every 48 h and transferred to 0.5 L plastic containers until larval hatching. Larvae were reared on an artificial diet until the second instar, at which point they were used for feeding assays with diets supplemented with soursop leaf extracts.

#### 4.4.2. Application of Extracts

Ethanolic leaf extracts of *A. muricata* were incorporated into an artificial diet and offered to second-instar *S. frugiperda* larvae following the methodology described previously [52,63] with slight modifications. For 500 g of diet, the formulation included agar (7 g), soy flour (40.5 g), brewer’s yeast (12.5 g), sugar (6.5 g), wheat germ (16.7 g), benzoic acid (0.8 g), Wesson’s salts (2.6 g), methyl paraben (0.5 g), ascorbic acid (2.15 g), vitamin mix (0.5 g), antibiotic (1 g), acetic acid 25% (6 mL), choline chloride 15% (3.6 mL), and distilled water (450 mL). Agar, soy flour, yeast, sugar, and wheat germ were mixed with water and autoclaved (121 °C, 20 min). Once cooled to 38–40 °C, the remaining ingredients were added and mixed thoroughly before use. Extracts from soursop plants inoculated with AMF consortia (CM and AD), AMF species (FM and RI), and non-mycorrhizal plants (SM) were dissolved in 1% Tween 80 solution at a concentration of 50 mg·mL^−1^ (20 mL) and mixed with 200 g of artificial diet, resulting in a final concentration of 0.25 mg extract·g^−1^ of diet. This concentration was selected by considering the extract yield and its practical feasibility for incorporation into the diet. A negative control consisting of 20 mL of 1% Tween 80 solution was included, as well as a positive control consisting of two concentrations of the commercial insecticide Palgus (Corteva Agriscience^®^, spinetoram 60 g a.i.·L^−1^, Wilmington, DE, USA). Palgus was dissolved in 1% Tween 80 at 0.066 or 0.66 µg a.i.·mL^−1^ (20 mL) and mixed with 200 g of artificial diet, resulting in final concentrations of 0.0066 or 0.066 µg·g^−1^ of diet.

Diet plugs of uniform size (≈1.8 g) were cut with a cork borer and placed individually in the wells of 12-well trays. One second-instar larva, starved for 6 h, was placed in each well. The trays containing insects were maintained in a growth chamber at 28 °C, 65% relative humidity, and a 12:12 h (light–dark) photoperiod. Larval survival was recorded every 24 h for 25 days. Surviving larvae were weighed on an analytical balance (BP121S, Sartorius, Göttingen, Germany) and transferred to fresh wells with an artificial diet until completing their development. For each surviving individual, the number of days from larva to pupa and from pupa to adult was recorded. Pupae were weighed using an analytical balance.

The experimental design was completely randomized, with 25 replicates per treatment (ethanolic extracts from plants inoculated with different AMF consortia, negative control, and positive control). Each replicate consisted of one well of a 12-well tray containing a diet plug from the corresponding treatment and a single *S. frugiperda* larva. No biological replication was performed in this assay; each larva was considered an independent experimental unit.

### 4.5. Determination of Total Annonacins in A. muricata Leaf Extracts

#### 4.5.1. Plant Extract

Leaves from mycorrhizal and non-mycorrhizal soursop plants were collected, frozen at −80 °C (Thermo Scientific^®^; model 900 series, Waltham, MA, USA), and lyophilized (Shin BioBase; model TFD5503, Dongducheon, Republic of Korea). Dried leaves were ground in a blender (Hamilton Beach^®^, Glen Allen, VA, USA), and 1 g of powder was mixed with 5 mL of methanol (99%) via sonication (Branson^®^ 2510; 40 kHz; 28–32 °C, Danbury, CT, USA) for 1 h. The samples were subsequently centrifuged for 30 min at 13,000 rpm, and the supernatant was evaporated in a Vacufuge Plus Eppendorf^®^ (45 °C) until the extract was obtained. The extraction procedure was repeated four times using different leaf samples.

#### 4.5.2. Annonacin Quantification via HPLC (González-López et al., 2025 [23])

*A. muricata* extracts were analyzed by means of high-performance liquid chromatography (HPLC) using a Waters Acquity UPLC H-Class system (Milford, MA, USA). Chromatographic separation was carried out at 30 °C on a Waters XSelect HSS C18 5 µm (4.6 mm × 150 mm) column (Waters, Milford, MA, USA). An isocratic system was used with acetonitrile–water (70:30) as the mobile phase, a flow rate of 1.20 mL·min^−1^, an injection volume of 15 µL, and UV detection at λ = 210 nm. The calibration curve (10–1000 µg·mL^−1^) was prepared using an annonacin standard (Cayman Chemical^®^, Ann Arbor, MI, USA; purity > 98%), yielding the regression equation y = 7012.5x + 35,643 (R^2^ = 0.9999). The limit of quantification (LOQ), defined as the lowest validated concentration with acceptable accuracy and precision, was 10 µg·mL^−1^. The limit of detection (LOD) was estimated as LOQ/3.3 = 3.0 µg·mL^−1^. A representative chromatogram is provided in Appendix A. Annonacin concentration in leaves was expressed as µg of annonacin per g of dry weight (µg·g^−1^ DW). For quantitative analysis, annonacin was extracted with methanol instead of ethanol because methanol shows better compatibility with the chromatographic system and ensures complete solubilization of the standard. This analytical extraction was used only for HPLC determination; in comparison, all biological assays were performed with ethanolic leaf extracts, as described in Section 4.2.

### 4.6. Statistical Analysis

All statistical analyses were performed in R Statistical Software v4.1.2 [64]. Data were first tested for normality using the Shapiro–Wilk test and for homogeneity of variances using Bartlett’s test. When assumptions of normality and homoscedasticity were not satisfied, non-parametric alternatives such as the Kruskal–Wallis and Dunn’s tests were applied. Regarding *S. frugiperda* and *T. pallidipennis*, survival data were analyzed using Kaplan–Meier survival curves, which were compared with the Log-Rank test (*p* ≤ 0.05) and adjusted using Holm–Šidák correction for multiple comparisons.

In *S. frugiperda*, larval weight after six days of feeding, larval-to-pupal duration, and pupal-to-adult duration were analyzed using the Kruskal–Wallis test (*p* ≤ 0.05), followed by Dunn’s multiple comparison test. Pupal weight was analyzed by means of one-way ANOVA followed by Tukey’s post hoc test (*p* ≤ 0.05). For *T. pallidipennis*, egg counts per female were analyzed using the Kruskal–Wallis test (*p* ≤ 0.05).

Finally, annonacin content in *A. muricata* leaf extracts was analyzed by means of one-way ANOVA, and mean separation was performed using the LSD test (*p* ≤ 0.05).

## 5. Conclusions

The results demonstrate that inoculation with arbuscular mycorrhizal fungi (AMF) modulates the insecticidal activity of *Annona muricata*, suggesting that mycorrhization may influence the synthesis or accumulation of bioactive secondary metabolites. AD extracts (annonacin = 1016 µg·g^−1^ DW) were most effective against *Triatoma pallidipennis*, whereas *Funneliformis mosseae* extracts achieved 72% larval mortality in *Spodoptera frugiperda*. These results suggest that acetogenins such as annonacin, present in ethanolic leaf extracts of mycorrhizal *A. muricata* plants, play a key role in their insecticidal activity. Overall, these findings highlight the potential of AMF-assisted cultivation to enhance the production of bioactive secondary metabolites with applications in sustainable pest management.

## Figures and Tables

**Figure 1 plants-14-03501-f001:**
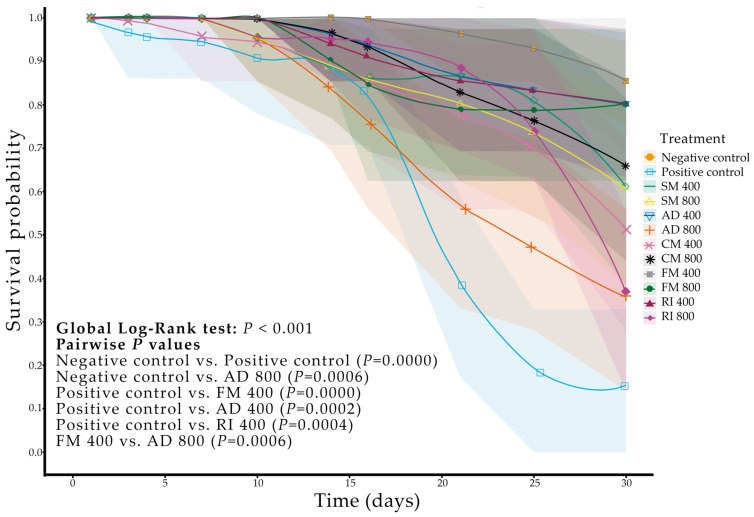
Survival of *Triatoma pallidipennis* adults exposed to ethanolic extracts from the leaves of *A. muricata* plants inoculated with arbuscular mycorrhizal fungi (AMF): Cerro del Metate (CM), Agua Dulce (AD), *Funneliformis mosseae* (FM), *Rhizophagus intraradices* (RI), without AMF (SM), DMSO 100% (Negative Control), and cypermethrin at 100 ng per insect (Positive Control). Sample size per treatment: 20 adults (10 females and 10 males). Global comparison by Log-Rank (Mantel–Cox): *p* < 0.001. Only significant pairwise comparisons are shown. Shaded areas represent 95% confidence interval (CI) bands.

**Figure 2 plants-14-03501-f002:**
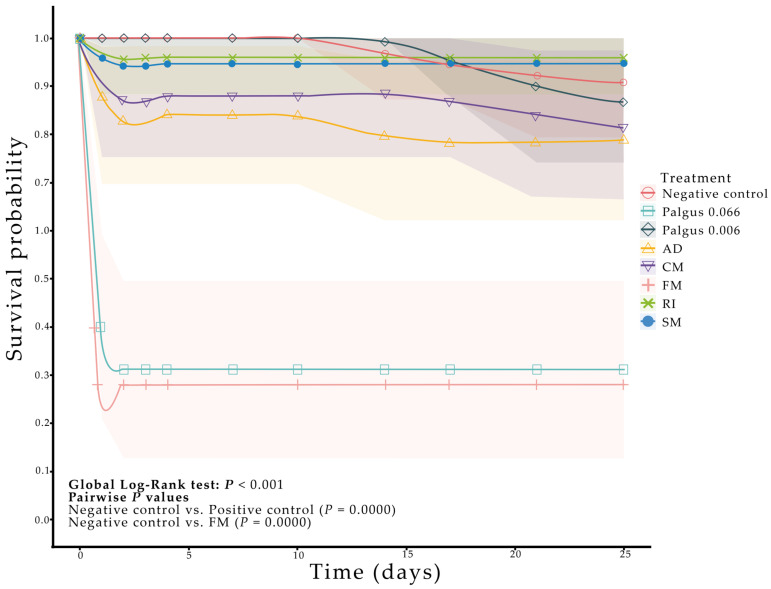
Survival of *Spodoptera frugiperda* larvae fed an artificial diet mixed with ethanolic extracts of *A. muricata* leaves inoculated with arbuscular mycorrhizal fungi (AMF). Treatments were extracts at a concentration of 0.25 mg·g^−1^ of soursop diet, colonized by the AMF: Cerro del Metate (CM), Agua Dulce (AD), *Funneliformis mosseae* (FM), *Rhizophagus intraradices* (RI), without AMF (SM), 1% Tween 80 (Negative control), Palgus insecticide at 0.0066 µg·g^−1^ of diet (Positive control 0.0066), and Palgus insecticide at 0.066 µg·g^−1^ of diet (Positive control 0.066). Sample size per treatment: 25 larvae (second instar). Global comparison by Log Rank (Mantel–Cox): *p* < 0.001. Only relevant pairwise comparisons are shown. Shaded areas represent 95% confidence interval (CI) bands.

**Table 1 plants-14-03501-t001:** Effect of a 30-day exposure to ethanolic leaf extracts of *Annona muricata* plants inoculated with arbuscular mycorrhizal fungi on mortality and egg number of *Triatoma pallidipennis* adults.

Treatment ^†^	Extract Concentration (mg·mL^−1^)	Mortality ^††^		Egg Number ^‡^		Annonacin Concentration ^‡‡^ (µg·g^−1^ DW)	Colonization ^§^ (%)
		n (♀ + ♂)	(%)	Mean (n)	Total (n)		
Negative control	--	3 (2 + 1)	15 ± 8.0	51.9 ± 35.9 (10)	519 (10)	--	--
Positive control	100 ng	17 (10 + 7)	85 ± 8.0 *	25.5 ± 12.7 (6)	307 (6)	--	--
AD	400	4 (2 + 1)	20 ± 8.9	43.9 ± 40.4 (10)	439 (10)	1016.0 ± 124.5 ab	7.6 ± 3.0 cd
	800	13 (5 + 7)	65 ± 10.6 *	42.3 ± 26.1 (9)	381 (9)		
CM	400	10 (4 + 5)	50 ± 11.2	54.3 ± 26.6 (8)	489 (8)	761.8 ± 154.3 ab	27.8 ± 5.7 ab
	800	7 (3 + 3)	35 ± 10.2	54.8 ± 35.6 (9)	494 (9)		
RI	400	5 (3 + 1)	25 ± 9.7	39.6 ± 27.1 (10)	396 (10)	1209.0 ± 272.7 a	41.0 ± 6.4 a
	800	13 (8 + 5)	65 ± 10.6 *	56.4 ± 34.4 (10)	564 (10)		
FM	400	3 (2 + 1)	15 ± 8.0	66.8 ± 34.0 (10)	602 (10)	997.6 ± 130.8 ab	18.0 ± 6.5 bc
	800	5 (3 + 2)	25 ± 9.7	39.6 ± 22.3 (10)	436 (10)		
SM	400	8 (2 + 5)	40 ± 11.0	53.8 ± 26.3 (9)	485 (9)	705.1 ± 47.5 b	2.2 ± 1.8 d
	800	8 (5 + 2)	40 ± 11.0	40.1 ± 23.4 (9)	361 (9)		

^†^ Treatments: Negative Control = 100% DMSO, Positive Control = 100 ng cypermethrin per insect, Agua Dulce (AD), Cerro del Metate (CM), *Rhizophagus intraradices* (RI), *Funneliformis mosseae* (FM), and non-mycorrhizal (SM). ^††^ Mortality percentage was calculated as %M = ((Total treated individuals − Dead individuals)/Total treated individuals) × 100. * Differences in mortality among treatments were analyzed using the Chi-square test (χ^2^), with significant differences (*p* < 0.05) relative to the negative control. Twenty triatomine adults were used per treatment (10 males + 10 females). Standard errors were estimated assuming a binomial distribution. ^‡^ Egg number was recorded 30 days after treating females with extracts and controls; values represent the mean ± SEM (standard errors). ^‡‡^ Values represent the mean ± SEM (n = 4). Different lowercase letters (a, b) indicate significant differences among treatments according to the LSD test (*p* ≤ 0.05; adapted from [23]). µg·g^−1^ DW = µg of annonacins per g of leaf dry weight. ^§^ Determined microscopically; data previously reported in [23]. Values represent the mean ± SEM; different lowercase letters (a, b, c, d) indicate significant differences among treatments according to the LSD test (*p* ≤ 0.05; adapted from [23]).

**Table 2 plants-14-03501-t002:** Effect of ethanolic leaf extracts from AMF-inoculated *Annona muricata* plants, incorporated into an artificial diet, on *Spodoptera frugiperda* larvae.

Treatments ^†^	Weight (mg)		Time (Days)		Mortality ^††^ (%)
	Larvae ^‡^ (n)	Pupa (n)	Larvae to Pupa	Pupa to Adult	
Negative control	11.3 ± 1.2 (22) a	227.7 ± 6.8 (19) b	19.0 ± 0.46 b	11.0 ± 0.28 a	12 ± 6.5
Positive control 0.0066	53.9 ± 4.2 (25) b	196.5 ± 5.3 (20) a	17.5 ± 0.55 c	11.7 ± 0.40 a	0
Positive control 0.066	0 ± 0 (0)	0 ± 0 (0)	0 ± 0	0 ± 0	100 ± 0 *
AD	9.0 ± 1.3 (15) a	228.9 ± 9.4 (14) b	20.1 ± 0.40 a	11.4 ± 0.28 a	40 ± 9.8
CM	8.2 ± 1.2 (19) a	231.8 ± 7.2 (16) b	17.8 ± 0.26 c	12.2 ± 0.28 a	24 ± 8.5
FM	44.1 ± 3.7 (7) b	204.4 ± 9.3 (6) ab	15.8 ± 0.47 d	12.3 ± 0.33 a	72 ± 9.0 *
RI	12.0 ± 2.1 (14) a	235.9 ± 9.0 (14) b	20 ± 0.40 a	10.1 ± 0.24 b	44 ± 9.9
SM	12.3 ± 1.2 (19) a	231.8 ± 4.2 (18) b	18.8 ± 0.30 ab	11.0 ± 0.27 a	24 ± 8.5

^†^ Treatments: Negative control = 1% Tween 80; Positive control = Palgus 0.0066 or 0.066 µg·g^−1^ of diet; extracts at a concentration of 0.25 mg·g^−1^ of diet from soursop plants colonized by AMF. ^‡^ Larval weight after six days of exposure to diets supplemented with the treatments. Values represent the mean ± SEM, and n indicates the number of individuals for which weight was recorded. ^††^ Mortality percentage was calculated as %M = ((Total treated individuals − Dead individuals)/Total treated individuals) × 100. * Differences in mortality among treatments were analyzed using the Chi-square test (χ^2^), with significant differences (*p* < 0.05) relative to the negative control. Standard errors were estimated assuming a binomial distribution. A total of 25 larvae were initially exposed per treatment. Different letters indicate significant differences according to Dunn’s test (*p* ≤ 0.05) for larval weight, larval-to-pupal duration, and pupal-to-adult duration. For pupal weight, Tukey’s test was used (*p* ≤ 0.05). Abbreviations: AD = Agua Dulce; CM = Cerro del Metate; FM = *Funneliformis mosseae*; RI = *Rhizophagus intraradices*; SM = non-mycorrhizal plants.

## Data Availability

The raw data supporting the conclusions of this article will be made available by the authors on request.

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
