# Peer review of "Arbuscular Mycorrhizal Fungi Enhance the Insecticidal Activity of Annona muricata L. Leaves"

_plants, 2025, doi:10.3390/plants14223501_

Round 1

Reviewer 1 Report

Comments and Suggestions for Authors

This manuscript explores the novel interaction between arbuscular mycorrhizal fungi (AMF) and the enhancement of secondary metabolites in Annona muricata, focusing on insecticidal effects against Spodoptera frugiperda and Triatoma pallidipennis. The topic is interesting, combining plant–microbe symbiosis with bioinsecticide potential. The experimental work is generally sound and clearly presented. However, the paper would benefit from improved organization, deeper discussion of mechanisms, and clarification of experimental details. Here are some comments.

Title and Abstract

Lines 1–3:

“Effect of arbuscular mycorrhizal fungi on the enhancement of the insecticidal activity of Annona muricata L. leaves.”

Clear and descriptive, but consider shortening: “Arbuscular mycorrhizal fungi enhance the insecticidal activity of Annona muricata L. leaves.” for smoother phrasing.

Lines 15–33 (Abstract):

  • Add 1 quantitative result (e.g., “FM treatment caused 72 % larval mortality”).
  • Clarify the specific AMF species linked with the highest activity (FM or AD).
  • Replace “These extracts also contained elevated concentrations of annonacin” → “These extracts showed significantly higher annonacin content (µg g⁻¹ DW).”
  • Grammar: “the production of secondary metabolites in medicinal plants” (delete “the”).
  • End abstract with a forward-looking statement: “This suggests potential for AMF-assisted cultivation to enhance botanical insecticides.”

Introduction

Lines 35–64:

  • The background on A. muricata phytochemistry is comprehensive but overly long. Consider merging Lines 47–64 into ~4 sentences focusing on acetogenins and their insecticidal role.
  • Line 53: “have largely been attributed to its wide diversity of acetogenins” → “are mainly attributed to acetogenins.”
  • Cite recent metabolomic studies (2023–2025) to replace older general citations [3, 9].

Lines 65–77:

  • Excellent transition to AMF; add 1 sentence clarifying the hypothesis:
    “We hypothesize that AMF symbiosis enhances acetogenin biosynthesis, increasing the insecticidal potential of A. muricata leaves.”
  • Typo: “how- ever” → “however”.
  • Clarify the novelty—AMF influence on insecticidal (not medicinal) metabolites has not been reported.

Results:

Lines 79–93:

  • State clearly the sample size: “Twenty adults per treatment (10 ♂ + 10 ♀)” should appear once in main text, not only in Table 1.
  • Lines 84–85: clarify “AD and RI at 800 mg mL⁻¹ achieved 65 % mortality” → “Extracts from AD and RI treatments (800 mg mL⁻¹) caused 65 % mortality.”
  • Explain whether sex affected survival (mention “no difference between ♂ and ♀”).

Table 1 & Lines 99–109:

  • Check unit consistency: use mg·mL⁻¹ throughout.
  • Provide SEM or SD for annonacin concentrations.
  • Clarify statistical letters (a, b, ab) in caption: what comparison test and n-value.

Figure 1 (Lines 111–115):

  • Add survival-curve legend with sample sizes.
  • Indicate whether differences were significant via Log-Rank test (add P values).

Lines 118–125:

  • Replace “showed significant differences” → “significantly reduced larval survival relative to the control (Log-Rank P ≤ 0.05).”
  • Clarify concentration of extract in diet (0.25 mg g⁻¹) within text for self-containment.

Lines 126–134:

  • The statement “no reduction in pupal biomass” contradicts later discussion—double-check consistency.
  • Line 130: express developmental time as mean ± SE days with statistical results (H value or F value).

Table 2 & Figure 2:

  • Include units for weight (mg) and time (days).
  • Ensure equal number formatting (e.g., “44.1 ± 7.0” not “44.1(7)”).
  • For clarity, move all abbreviations (AD, CM, FM, RI, SM) below table.

Discussion

Lines 153–180

  • The discussion repeats literature; instead, emphasize how AMF-mediated annonacin increase explains observed mortality.
  • Line 175: specify the annonacin concentration range that correlated with mortality.
  • Suggest adding a short mechanistic paragraph on nutrient-mediated secondary metabolism induction by AMF (P- and N-uptake link).

Lines 182–214

  • Well-referenced but needs stronger synthesis. Explain why Funneliformis mosseae (FM) produced higher toxicity—possible species-specific signaling?
  • Lines 198–205: clarify that increased larval weight may indicate compensatory feeding (hormesis), not positive health.
  • End discussion with a limitation: absence of full metabolomic profiling; annonacin may not be sole active compound.

Materials and Methods

Lines 217–232

  • Include pot replication details in one concise sentence.
  • Add light intensity and temperature range for greenhouse conditions.

Lines 233–245

  • Report extraction yield (% w/w) and whether solvent was completely evaporated.
  • Replace “Branson 2510; 5 Hz; 28–32 °C” → “Branson 2510; 40 kHz; 28–32 °C” (5 Hz is likely a typo).

Lines 246–267

  • Confirm ethical approval or institutional authorization for animal use (NZW rabbits).
  • Add statement on randomization of insects across treatments.

Lines 285–307

  • Clarify diet composition and whether it was autoclaved.
  • Add number of larvae per treatment and repetition (biological vs technical).

Lines 316–337

  • Include calibration curve equation, detection limit (LOD/LOQ), and R² value.
  • Add representative chromatogram to Supplementary Data.

Conclusions

  • Improve specificity: “AD extracts (annonacin ≈ 1016 µg g⁻¹ DW) were most effective against T. pallidipennis, whereas FM extracts achieved 72 % larval mortality in S. frugiperda.
  • End with perspective: “These findings highlight the potential of AMF-assisted cultivation for producing bioactive secondary metabolites with pest-management applications.”

Language and Formatting

  • Italicize all species names.
  • Unify decimal markers (use “.” not “, ”).
  • Replace “µg·g⁻¹ DW” consistently (avoid alternation with “µg g⁻¹ DW”).
  • Proofread minor grammar (articles, plural consistency).

Author Response

Reviewer 1

Comments and Suggestions for Authors

This manuscript explores the novel interaction between arbuscular mycorrhizal fungi (AMF) and the enhancement of secondary metabolites in Annona muricata, focusing on insecticidal effects against Spodoptera frugiperda and Triatoma pallidipennis. The topic is interesting, combining plant–microbe symbiosis with bioinsecticide potential. The experimental work is generally sound and clearly presented. However, the paper would benefit from improved organization, deeper discussion of mechanisms, and clarification of experimental details. Here are some comments.

Title and Abstract

Lines 1–3:

Comment 1: “Effect of arbuscular mycorrhizal fungi on the enhancement of the insecticidal activity of Annona muricata L. leaves.”

  1. Clear and descriptive, but consider shortening: “Arbuscular mycorrhizal fungi enhance the insecticidal activity of Annona muricata L. leaves.” for smoother phrasing.

Response: We appreciate the reviewer’s suggestion. We agree that the revised title is more concise and maintains the original meaning. Therefore, we have changed the title to: “Arbuscular mycorrhizal fungi enhance the insecticidal activity of Annona muricata L. leaves.”

Lines 15–33 (Abstract):

  1. Add 1 quantitative result (e.g., “FM treatment caused 72 % larval mortality”).

Response: To strengthen the abstract with quantitative information, we incorporated a representative numerical result from our assays. Specifically, we added that Funneliformis mosseae treatment caused 72 % larval mortality in Spodoptera frugiperda.

  1. Clarify the specific AMF species linked with the highest activity (FM or AD).

Response: We appreciate the reviewer’s insightful comment. To address this, we clarified in the abstract which AMF treatment exhibited the highest insecticidal activity. Specifically, we now indicate that Funneliformis mosseae caused 72 % larval mortality in Spodoptera frugiperda, while the Agua Dulce consortium showed the strongest activity against Triatoma pallidipennis (65 % mortality). This revision provides clearer information on the specific AMF species and consortium associated with the greatest effects.

  1. Replace “These extracts also contained elevated concentrations of annonacin” → “These extracts showed significantly higher annonacin content (µg g⁻¹ DW).”

Response: We have revised the sentence as suggested to improve precision and clarity. The phrase “These extracts also contained elevated concentrations of annonacin” was replaced with “These extracts showed significantly higher annonacin content (µg g⁻¹ DW)” to better reflect the statistical significance and to include the measurement units.

  1. Grammar: “the production of secondary metabolites in medicinal plants” (delete “the”).

Response: We have removed the article “the” before “production” to correct the sentence, which now reads: “production of secondary metabolites in medicinal plants.”

  1. End abstract with a forward-looking statement: “This suggests potential for AMF-assisted cultivation to enhance botanical insecticides.”

Response: We appreciate the reviewer’s recommendation. We have incorporated the suggested forward-looking statement at the end of the abstract to highlight the broader implication of our findings.

Introduction

Lines 35–64:

  1. The background on  muricataphytochemistry is comprehensive but overly long. Consider merging Lines 47–64 into ~4 sentences focusing on acetogenins and their insecticidal role.

Response: We thank the reviewer for this valuable suggestion. Following the recommendation, we revised the section on A. muricata phytochemistry to make it more concise and focused on acetogenins and their insecticidal role

  1. Line 53: “have largely been attributed to its wide diversity of acetogenins” → “are mainly attributed to acetogenins.”

Response: We appreciate the reviewer’s suggestion. However, the specific phrase “have largely been attributed to its wide diversity of acetogenins” was removed during the reduction of the paragraph to comply with the request for conciseness and focus.

  1. Cite recent metabolomic studies (2023–2025) to replace older general citations [3, 9].

Response: We appreciate the reviewer’s suggestion to include more recent metabolomic references. We have replaced the older general citations [3, 9] with up-to-date studies (2023–2025) that apply advanced metabolomic techniques to Annona muricata extracts, including UPLC–ESI–MS/MS profiling (e.g. Abdallah et al. 2024) which identify acetogenins, phenolics, and alkaloids in leaf extracts

Lines 65–77:

  1. Excellent transition to AMF; add 1 sentence clarifying the hypothesis:
    “We hypothesize that AMF symbiosis enhances acetogenin biosynthesis, increasing the insecticidal potential of A. muricata leaves.”

Response: Following the recommendation, we added a clear statement of the study hypothesis at the end of the introductory section.

  1. Typo: “how- ever” → “however”.

Response: We appreciate the reviewer’s observation. The word separation (“how- ever”) from the automatic hyphenation settings embedded in the official Plants template, which applies justified text formatting with automatic line breaks. The text has been carefully reviewed, and all instances of incorrect separation were verified to be formatting effects rather than spelling errors. Therefore, the original hyphenation has been retained to maintain consistency with the journal’s template.

  1. Clarify the novelty—AMF influence on insecticidal(not medicinal) metabolites has not been reported.

Response: We thank the reviewer for this helpful comment. To emphasize the novelty of our study, we added a sentence in the introduction stating that previous research has focused mainly on the effect of AMF on medicinal or pharmacologically active molecules, whereas their influence on insecticidal compounds remains unexplored.

Results:

Lines 79–93:

  1. State clearly the sample size: “Twenty adults per treatment (10 ♂ + 10 ♀)” should appear once in main text, not only in Table 1.

Response: We have now included the sample size directly in the main text of the Materials and Methods section (subsection 4.3.2). The revised text reads: “Twenty triatomine adults were used per treatment (10 males and 10 females).” This addition ensures that the sample size is clearly stated in the main text, as requested.

  1. Lines 84–85: clarify “AD and RI at 800 mg mL⁻¹ achieved 65 % mortality” → “Extracts from AD and RI treatments (800 mg mL⁻¹) caused 65 % mortality.”

Response: We thank the reviewer for this helpful suggestion. We have revised the sentence for clarity as recommended. The corrected version now reads: “Extracts from AD and RI treatments (800 mg·mL⁻¹) caused 65% mortality.” This modification improves readability and precision in describing the result.

  1. Explain whether sex affected survival (mention “no difference between ♂ and ♀”).

Response: We thank the reviewer for this comment. No apparent differences in survival were observed between males and females within the same treatment; therefore, sex-based statistical comparisons were not performed, and data from both sexes were pooled for the final analysis. This clarification has been added to the Results section.

Table 1 & Lines 99–109:

  1. Check unit consistency: use mg·mL⁻¹ throughout.

Response: We have carefully revised the entire manuscript to ensure consistency in all units. All concentrations are now expressed as mg·mL⁻¹ and µg·g⁻¹ throughout the text, tables, and figure captions.

  1. Provide SEM or SD for annonacin concentrations.
  2. Clarify statistical letters (a, b, ab) in caption: what comparison test and n-value.

Response (17 and 18): We have revised the caption of Table 1 to clarify the meaning of the statistical letters. The caption now specifies that values are presented as mean ± SEM (n = 4) and that different lowercase letters (a, b) indicate significant differences among treatments for annonacin concentration according to the LSD test (P ≤ 0.05).

Figure 1 (Lines 111–115):

  1. Add survival-curve legend with sample sizes.

Response: We have added the sample size to the legends of Figures 1. The revised legend now specify the number of insects used per treatment (n = 20 adults for T. pallidipennis).

  1. Indicate whether differences were significant via Log-Rank test (add P values).

Response: We appreciate the reviewer’s suggestion. The Results section and Figure 1 have been updated to indicate that survival differences among treatments were analyzed using the Log-Rank (Mantel–Cox) test. The global test result (P < 0.001) is now reported in Figure 1 and Figure 2, and significant pairwise P-values between treatments have been added to highlight statistically meaningful differences. However, in Figure 2 only the relevant and significant P-values were added to avoid saturation. All P-values obtained from the analysis are presented in the following figure for completeness. To improve clarity and visual consistency, very small P-values were rounded in Figure 1.

Figure 1. P values of Log-Rank (Mantel–Cox) test in T. pallidipennis.  

Figure 2. P values of Log-Rank (Mantel–Cox) test in S. frugiperda.  

Lines 118–125:

  1. Replace “showed significant differences” → “significantly reduced larval survival relative to the control (Log-Rank P ≤ 0.05).”

Response: It now reads: “Survival curve analysis of S. frugiperda larvae exposed to soursop extract incorporated into the diet at 0.25 mg·g⁻¹ revealed that both the FM treatment and Palgus 0.066 µg·g⁻¹ (positive control) significantly reduced larval survival relative to the negative control (1% Tween 80; Log-Rank, P ≤ 0.05; Table 2; Figure 2).”

  1. Clarify concentration of extract in diet (0.25 mg g⁻¹) within text for self-containment.

Response: We have added the concentration of the soursop extract in the diet (0.25 mg·g⁻¹) directly within the text to make the paragraph self-contained and consistent with the experimental details reported in the Methods section.

Lines 126–134:

  1. The statement “no reduction in pupal biomass” contradicts later discussion—double-check consistency.

Response: We thank the reviewer for pointing out this inconsistency. The Results section was revised to clarify that a reduction in pupal biomass occurred only in the synthetic insecticide treatment (Palgus 0.0066 µg·g⁻¹), while A. muricata extracts did not reduce pupal biomass relative to the negative control. The paragraph now reads:

“After six days of feeding, larvae treated with Palgus 0.066 µg·g⁻¹ and FM showed significantly higher weight compared with the negative control (Dunn, p ≤ 0.05; Table 2). At the pupal stage, individuals exposed to the lower Palgus dose (0.0066 µg·g⁻¹) exhibited reduced biomass (196.6 ± 5.3 mg) relative to the other treatments (Table 2). In contrast, pupal biomass of larvae treated with A. muricata extracts did not differ from the negative control, indicating that the extracts did not reduce pupal weight compared with untreated larvae.”

In addition, the Discussion section was revised for consistency. The statement was rephrased to clearly indicate that A. muricata extracts did not affect pupal biomass compared with the negative control, whereas the synthetic insecticide significantly reduced pupal weight. The revised text reads:

“At the pupal stage, biomass was unaffected by A. muricata extracts, in contrast to the synthetic insecticide (Palgus 0.0066 µg·g⁻¹), which significantly reduced pupal weight.”

  1. Line 130: express developmental time as mean ± SE days with statistical results (H value or F value).

Response: We thank the reviewer for this valuable suggestion. The section on developmental time has been revised to include both the descriptive values (mean ± SE days) and the corresponding statistical results from the Kruskal–Wallis test. The text now reads: “Regarding developmental time, negative control larvae reached pupation in 19.0 ± 0.46 days, whereas larvae exposed to Palgus 0.066 µg·g-1 and FM developed faster (17.5 ± 0.55 and 15.8 ± 0.47 days, respectively). Conversely, larvae treated with AD and RI extracts exhibited longer developmental times (20.1 ± 0.40 and 20.0 ± 0.39 days, respectively). Significant differences among treatments were confirmed by the Kruskal–Wallis test (H = 48.23, P < 0.0001). For pupal duration, significant differences among treatments were observed (H = 19.86, P = 0.0029). Only the RI treatment differed significantly from the others, showing a shorter pupal stage (10.1 ± 0.24 days) (Table 2).”

Table 2 & Figure 2:

  1. Include units for weight (mg) and time (days).

Response: We appreciate the reviewer’s observation. Units for weight (mg) and time (days) have been added in Table 2 and in the corresponding text to ensure clarity and consistency with the reported variables.

  1. Ensure equal number formatting (e.g., “44.1 ± 7.0” not “44.1(7)”).

Response:  We appreciate the reviewer’s observation. The numerical formatting in Tables 1 and 2 has been standardized for consistency. All values are now expressed as mean ± standard error (SE), followed by the sample size in parentheses (e.g., 44.1 ± 7.0 (7)). The number of decimal places was also harmonized within each column for clarity.

In addition, we note that the standard error was not included for the number of dead individuals and the total number of eggs, since these are absolute counts and therefore not subject to variation measures such as the SE.

  1. For clarity, move all abbreviations (AD, CM, FM, RI, SM) below table.

Response: We have moved the abbreviations (AD, CM, FM, RI, SM) to a separate line below the table for greater clarity and readability, as recommended.

Discussion

Lines 153–180

  1. The discussion repeats literature; instead, emphasize how AMF-mediated annonacin increase explains observed mortality.

Response: We appreciate the reviewer’s suggestion. The discussion section was revised to reduce redundant literature and to better highlight the mechanistic link between AMF inoculation, annonacin accumulation, and the observed mortality in T. pallidipennis. The revised text now explains that annonacin acts as a mitochondrial complex I inhibitor, disrupting ATP synthesis and leading to insect mortality (González-Coloma et al., 2002). Furthermore, we integrated a mechanistic paragraph describing how AMF-mediated nutrient uptake (particularly P and N) enhances primary metabolic flux through the MVA and MEP pathways, promoting the biosynthesis of terpenoid-derived compounds such as annonacin. This modification establishes a clearer causal connection between AMF colonization and the increased insecticidal activity observed in our study.

  1. Line 175: specify the annonacin concentration range that correlated with mortality.

Response: We thank the reviewer for this useful observation. The Discussion section (lines 175–177) was modified to specify the annonacin concentration range associated with the highest adult mortality. The revised text now states that the highest mortality rates (65–72%) were observed in treatments containing annonacin concentrations between 998 and 1209 µg·g⁻¹ DW (FM and RI treatments, respectively).

  1. Suggest adding a short mechanistic paragraph on nutrient-mediated secondary metabolism induction by AMF (P- and N-uptake link).

Response: We appreciate the reviewer’s valuable suggestion. A new paragraph was added to the Discussion section to describe the nutrient-mediated mechanisms by which AMF can induce secondary metabolism (211-219).

Lines 182–214

  1. Well-referenced but needs stronger synthesis. Explain why Funneliformis mosseae(FM) produced higher toxicity—possible species-specific signaling?

Response: We appreciate the reviewer’s insightful comment. To address this point, we incorporated a new paragraph in the Discussion section explaining the potential mechanisms underlying the higher toxicity observed in F. mosseae (FM)-inoculated plants. The revised paragraph discusses that F. mosseae has been frequently reported to enhance the accumulation of secondary metabolites in various plant species, possibly through nutrient-mediated and signaling-related mechanisms involving jasmonate and phenylpropanoid pathways (Rivero et al., 2015; Amani Machiani et al., 2022; Rahmat and Soheilikhah, 2024). We also mention that few studies suggest differential effects among AMF species on host metabolite profiles (Zubek et al., 2012; Copetta et al., 2006; Khaosaad et al., 2006), supporting the possibility of species-specific interactions. The paragraph concludes by emphasizing that further metabolomic and signaling studies are required to confirm these hypotheses.

  1. Lines 198–205: clarify that increased larval weight may indicate compensatory feeding (hormesis), not positive health.

Response: We appreciate the reviewer’s insightful comment. The discussion has been revised to clarify that the increase in larval biomass should not be interpreted as improved larval health but rather as a potential compensatory or hormetic response to dietary toxicity. This clarification is now explicitly stated in the text, and the revised paragraph includes supporting evidence on hormetic effects induced by sublethal insecticide concentrations (Calabrese & Baldwin, 2002) and AMF-related metabolic modulation (Liu et al., 2025).

  1. End discussion with a limitation: absence of full metabolomic profiling; annonacin may not be sole active compound.

Response: We appreciate this suggestion. A new paragraph was added at the end of the Discussion to acknowledge this limitation. The revised text now highlights that the lack of full metabolomic profiling restricts the identification of additional active compounds and that annonacin may not be the sole metabolite responsible for the observed bioactivity.

Materials and Methods

Lines 217–232

  1. Include pot replication details in one concise sentence.

Response: We have condensed the description of the experimental replication into a single, concise sentence. The revised text now reads:

“The experiment consisted of five treatments (two AMF consortia, two AMF species, and a non-mycorrhizal control) arranged in a completely randomized design with seven replicates per treatment, each replicate corresponding to one pot containing a single soursop plant.”

  1. Add light intensity and temperature range for greenhouse conditions.

Response: We have added detailed greenhouse conditions, including temperature, relative humidity, and photosynthetically active radiation (PAR), as follows:

“Inoculated plants were maintained under greenhouse conditions (27.8 ± 0.26 °C morning; 34 ± 0.15 °C afternoon; 56–41 % relative humidity; 56.5–22.2 µmol m⁻² s⁻¹ PAR) for 19 months and watered weekly with tap water.”

Lines 233–245

  1. Report extraction yield (% w/w) and whether solvent was completely evaporated.

Response: We have added the extraction yield and specified that ethanol was completely evaporated. The revised text now reads:

“... evaporated in a Vacufuge Plus Eppendorf® concentrator at 45 °C until complete removal of the solvent. The resulting crude extract was stored at 4 °C, protected from light, until use in the bioassays. Extraction yield averaged 25.7 ± 3.4 % (w/w) based on dry leaf weight.”

  1. Replace “Branson 2510; 5 Hz; 28–32 °C” → “Branson 2510; 40 kHz; 28–32 °C” (5 Hz is likely a typo).

Response: We appreciate the reviewer’s careful observation. The value “5 Hz” has been corrected to “40 kHz.”

Lines 246–267

  1. Confirm ethical approval or institutional authorization for animal use (NZW rabbits).

Response: Response: We appreciate the reviewer’s observation. Ethical approval for the use of T. pallidipennis and NZW rabbits was granted by the Ethics Committee for Research (Comité de Ética en Investigación), Coordination of Research and Postgraduate Studies, Centro Universitario del Sur, University of Guadalajara (approval no. CIP/CEI/2024/20). This information has been added to the Materials and Methods section.

  1. Add statement on randomization of insects across treatments.

Response: We thank the reviewer for this important suggestion. A statement describing the randomization procedure used to distribute the insects among treatments has been added to subsection 4.3.2 (Application of extracts).

Lines 285–307

  1. Clarify diet composition and whether it was autoclaved.

Response: We have added details on the artificial diet composition and preparation, specifying the ingredients, quantities, and sterilization process.

  1. Add number of larvae per treatment and repetition (biological vs technical).

Response: We thank the reviewer for this suggestion. The number of larvae and the type of replicates have been clarified in the Materials and Methods section. The following statement has been added:
“The experimental design was completely randomized with 25 replicates per treatment (ethanolic extracts with different AMF inocula, negative control, and positive control). Each replicate consisted of one well of a 12-well tray containing a diet plug from the corresponding treatment and a single S. frugiperda larva. No biological replication was performed in this assay.”

Lines 316–337

  1. Include calibration curve equation, detection limit (LOD/LOQ), and R² value.\

Response: We have revised the HPLC section to include the calibration curve equation, the determination coefficient (R²), and the limits of detection and quantification. The calibration curve (10–1000 µg·mL⁻¹) yielded the regression equation y = 7012.5x + 35643 (R² = 0.9999). The limit of quantification (LOQ), defined as the lowest validated concentration with acceptable accuracy and precision, was 10 µg·mL⁻¹, and the limit of detection (LOD) was estimated as LOQ/3.3 = 3.0 µg·mL⁻¹.

  1. Add representative chromatogram to Supplementary Data.

Response: As requested, we have added a representative chromatogram of the annonacin standard to the Supplementary Materials section. The chromatogram shows a retention time (tᵣ) of 18.06 min and detection at 210 nm under the reported chromatographic conditions.

Conclusions

  1. Improve specificity: “AD extracts (annonacin ≈ 1016 µg g⁻¹ DW) were most effective against T. pallidipennis, whereas FM extracts achieved 72 % larval mortality in  frugiperda.

Response: We revised the Conclusions section to include specific quantitative data that support the main findings. The revised text now specifies the concentration of annonacin in AD extracts and the percentage of larval mortality observed with FM extracts, thus improving the precision and interpretability of the results.

  1. End with perspective: “These findings highlight the potential of AMF-assisted cultivation for producing bioactive secondary metabolites with pest-management applications.”

Response: We modified the final paragraph of the Conclusions to include a broader perspective that emphasizes the potential of arbuscular mycorrhizal fungi (AMF) in enhancing the production of bioactive secondary metabolites for sustainable pest management applications.

Language and Formatting

  1. Italicize all species names.
  1. Unify decimal markers (use “.” not “, ”).
  2. Replace “µg·g⁻¹ DW” consistently (avoid alternation with “µg g⁻¹ DW”).
  3. Proofread minor grammar (articles, plural consistency).

Response: All suggested editorial corrections have been implemented throughout the manuscript. Specifically, all scientific species names were italicized; decimal markers were standardized to periods (“.”) instead of commas; the unit “µg·g⁻¹ DW” was used consistently across the text, tables, and figures; and minor grammatical issues, including article usage and plural consistency, were carefully proofread and corrected.

Reviewer 2 Report

Comments and Suggestions for Authors
  1. Line 75-77: "one of the three main Mexican kissing bugs responsible for transmitting Trypanosoma cruzi, the causative agent of Chagas disease [27].":

Delete it or merge with another paragraph

  1. Results:

The authors must add photos for the fall armyworm Spodoptera frugiperda and T. pallidipennis after treatment with extracts

  1. The discussion is poorly and need to improve it

  1. Materials and Methods:

For Section: Statistical analysis:

Merge three sections of "Statistical analysis"  in one sections and be in a last section in materials and methods

  1. The section:"4.1. Plant material and mycorrhizal inoculation":

What is the resource for seeds and any species of AMF as Funneliformis mosseae (FM) and Rhizophagus intraradices (RI)? What are the species in "The consortia contain several AMF species, "? What is the resource?

  1. The conclusions are poor;y and need to rewrite without repeating the results and discussion and further studies.

Author Response

Reviewer 2

  1. Line 75-77: "one of the three main Mexican kissing bugs responsible for transmitting Trypanosoma cruzi, the causative agent of Chagas disease [27].":

Delete it or merge with another paragraph

Response: We appreciate the suggestion. The phrase describing T. pallidipennis as one of the main Mexican kissing bugs transmitting Trypanosoma cruzi was removed to improve the flow and conciseness of the paragraph. The sentence now simply refers to T. pallidipennis as “the kissing bug Triatoma pallidipennis.

  1. Results:
  1. The authors must add photos for the fall armyworm Spodoptera frugiperda and pallidipennis after treatment with extracts

Response: Photographs showing Spodoptera frugiperda larvae and Triatoma pallidipennis adults during the experimental setup and treatment with the ethanolic extracts have been included as a new supplementary figure (Figure S2). Although formal photographic documentation for all treatments was not available, the supplied images provide a representative view of the experimental procedures and insect handling under the described conditions. These photos help illustrate the context of the bioassays and complement the methodological description in the manuscript.

  1. The discussion is poorly and need to improve it

Response: We have carefully revised the Discussion section to improve its clarity, structure, and analytical depth. Repetitive statements were removed, and each paragraph was reorganized to better connect our findings with previous studies. In particular, we clarified the mechanistic link between AMF inoculation, nutrient uptake, and the observed increase in annonacin-related insecticidal activity. We also included a concise comparative context with other AMF–plant systems (e.g., Mimosa tenuiflora, Perilla frutescens, and Trifolium pratense) to situate our results within a broader ecological framework.

  1. Materials and Methods:

For Section: Statistical analysis:

  1. Merge three sections of "Statistical analysis"  in one sections and be in a last section in materials and methods

Response: We appreciate the suggestion. The three separate “Statistical analysis” sections have been merged into a single subsection placed at the end of the Materials and Methods section. This unified section now includes all the statistical tests used for S. frugiperda, T. pallidipennis, and annonacin quantification, improving the structure and readability of the manuscript.

  1. The section:"4.1. Plant material and mycorrhizal inoculation":

What is the resource for seeds and any species of AMF as Funneliformis mosseae (FM) and Rhizophagus intraradices (RI)? What are the species in "The consortia contain several AMF species, "? What is the resource?

Response: We thank the reviewer for this observation. The Materials and Methods section (4.1. Plant material and mycorrhizal inoculation) was revised to specify the sources of the plant material and AMF inocula. Seeds of Annona muricata were obtained from fresh fruits collected in the main soursop-producing region of Compostela, Nayarit, Mexico.

Spores of Funneliformis mosseae were obtained from monosporal propagation in trap pots containing Sorghum bicolor, Tagetes erecta, and Medicago sativa. This isolate originated from the Cerro del Metate site in Tzitzio, Michoacán, and is deposited at the National Genetic Resources Center (Centro Nacional de Recursos Genéticos, CNRG-INIFAP; Tepatitlán, Jalisco, Mexico) under the strain name QR01 and accession code CM-CNRG-TB233 (document below).

Spores of Rhizophagus intraradices were obtained from propagation pots containing Sorghum bicolor, Tagetes erecta, and Medicago sativa, using a commercial inoculum known as Micorriza INIFAP (INIFAP, 2016).

Additionally, the AMF consortia Cerro del Metate (CM) and Agua Dulce (AD) were collected from the rhizosphere of Agave cupreata plants growing in natural and cultivated sites in Michoacán, Mexico, respectively. The CM consortium comprises 13 AMF species, and the AD consortium contains nine species, as detailed in Trinidad-Cruz et al. (2017a). Both consortia are preserved in the Phytopathology Laboratory of the Plant Biotechnology Division at CIATEJ.

  1. The conclusions are poor; and need to rewrite without repeating the results and discussion and further studies.

Response: We appreciate the reviewer’s suggestion. The Conclusions section was completely revised to improve clarity, structure, and scientific focus. The new version avoids redundancy with the Discussion and emphasizes the main findings in a concise manner. Quantitative data (annonacin concentration and mortality percentages) were retained only to comply with the recommendations of other reviewers, who requested the inclusion of specific numerical results to strengthen the interpretation. The revised conclusion now highlights the main outcomes and the broader relevance of AMF-assisted cultivation for enhancing the production of bioactive secondary metabolites with potential applications in sustainable pest management.

Reviewer 3 Report

Comments and Suggestions for Authors

Comments:

Reduce textual similarity (42%) – Review the manuscript to minimize similarity with previously published work.

In the Abstract and Introduction, the following clarifications are required:

(Q1) Abstract – Clarify which extracts (AD, FM, RI, etc.) showed the highest efficacy for each insect species, and provide the exact mortality percentages to support these statements.

(Q2) Introduction – Knowledge gaps – Emphasize more clearly the existing gaps in knowledge. Specifically, highlight that the interaction between arbuscular mycorrhizal fungi (AMF), the production of secondary metabolites, and their insecticidal potential remains poorly understood.

(Q3) Introduction – Rationale for insect models – Justify the dual choice of Triatoma pallidipennis (of medical importance) and Spodoptera frugiperda (a major agricultural pest). Explain how this combination strengthens the relevance of the study by linking human health with agricultural challenges.

In the section Materials & Methods, the following clarifications should be provided:

(Q1) Inoculation protocol – Justify the choice of using 100 spores per plant. Indicate whether this inoculum level was determined from preliminary assays, drawn from published literature, or based on standard practice in similar studies.

(Q2) Extract concentrations – Clarify the rationale for applying concentrations of 400–800 mg/mL and 0.25 mg/g diet. Specify whether these values were selected according to preliminary dose–response experiments, data from literature, or practical considerations.

(Q3) AMF colonization – Provide information on how the establishment of arbuscular mycorrhizal fungi (AMF) was confirmed. State whether root colonization was evaluated microscopically (e.g., percentage of colonized root length, staining method), as this validation is essential to confirm effective AMF symbiosis.

(Q4) Statistical analysis and ethics – Indicate whether the assumptions of parametric statistical tests (normality, homogeneity of variances) were verified and whether non-parametric alternatives were used when assumptions were not satisfied. In addition, include the details of ethical approval for the use of rabbits in Triatoma pallidipennis feeding experiments, specifying the approving committee, protocol number, and compliance with animal welfare standards.

In the section Results, the following points should be addressed:

(Q1) Tables – While the tables are clear, improvements are needed. In Table 1 (mortality/eggs/annonacin), standard errors or 95% confidence intervals should be added to mortality values. In Table 2 (larval data), mortality is currently presented as percentages only; including the raw number of individuals per treatment would enhance transparency and reproducibility.

(Q2) Figures – For Figures 1 and 2, survival curves should be presented with error shading or confidence interval bands, rather than only step curves, to provide a better representation of variability and statistical reliability.

(Q3) Results text – The narrative description of results should not only state that treatments were "effective," but explicitly highlight which treatments differed significantly, supported by statistical comparisons.

(Q4) Annonacin quantification – Although annonacin quantification appears in Table 1, it is not clearly emphasized in the text. A dedicated subsection should be created in the Results to summarize the HPLC quantification of annonacin and to link these findings with the observed bioassay outcomes.

(Q5) Minor typos / taxonomic clarification – Double-check the species name in Table 1. The text currently refers to Meccus pallidipennis, but the correct and accepted name is Triatoma pallidipennis.

In the section Discussion, the following clarifications are required:

(Q1) Larval biomass paradox – The observed paradoxical increase in larval biomass in some treatments should be clarified. At present, the explanation remains speculative; a more detailed interpretation or possible mechanistic hypotheses are needed.

(Q2) Potential applications – Expand the discussion on the practical implications of combining AMF and Annona extracts. Specifically, evaluate whether such approaches could replace or complement synthetic insecticides, and acknowledge limitations such as the potential toxicity of annonacins, challenges in extraction scalability, and the need for field-level validation.

(Q3) Comparative context – Include a concise comparison with other plant–AMF systems that are reported to enhance insecticidal secondary metabolites. This would help situate the findings within a broader ecological and applied framework.

In the section References, the following clarification is required:

(Q1) References formatting – Ensure consistency in the formatting of references. In particular, all species names should be italicized throughout the reference list, following standard taxonomic conventions. Verify that journal names, volume/issue numbers, and page ranges are also presented uniformly.

Author Response

Reviewer 3

Comments:

  1. Reduce textual similarity (42%) – Review the manuscript to minimize similarity with previously published work.

Response: We appreciate the reviewer’s suggestion. The manuscript has been thoroughly restructured to minimize textual similarity and enhance its originality and clarity.

In the Abstract and Introduction, the following clarifications are required:

  1. (Q1) Abstract – Clarify which extracts (AD, FM, RI, etc.) showed the highest efficacy for each insect species, and provide the exact mortality percentages to support these statements.

Response: We appreciate the reviewer’s comment. The Abstract was revised to specify which extracts showed the highest efficacy against each insect species and to include the corresponding mortality percentages.

  1. (Q2) Introduction – Knowledge gaps – Emphasize more clearly the existing gaps in knowledge. Specifically, highlight that the interaction between arbuscular mycorrhizal fungi (AMF), the production of secondary metabolites, and their insecticidal potential remains poorly understood.

Response: We appreciate this valuable comment. The final paragraph of the Introduction was revised to more clearly emphasize the existing knowledge gap. This clarification reinforces the novelty and rationale of our study. “However, despite the extensive evidence of AMF-mediated enhancement of bioactive compounds, little is known about how these symbiotic associations affect the synthesis of insecticidal metabolites. The interaction between AMF colonization, the regulation of secondary metabolite pathways, and the resulting insecticidal potential of plants remains scarcely explored.”

  1. (Q3) Introduction – Rationale for insect models – Justify the dual choice of Triatoma pallidipennis (of medical importance) and Spodoptera frugiperda (a major agricultural pest). Explain how this combination strengthens the relevance of the study by linking human health with agricultural challenges.

Response: The selection of these insect models was based on their contrasting ecological and health relevance. S. frugiperda is one of the most destructive agricultural pests worldwide, causing severe losses in maize and other crops, while T. pallidipennis is a medically important vector of Trypanosoma cruzi, the etiological agent of Chagas disease. Evaluating both species provides a broader perspective on the potential of AMF-enhanced plant metabolites as bioinsecticidal agents effective against pests that threaten both crop productivity and human health.

In the section Materials & Methods, the following clarifications should be provided:

  1. (Q1) Inoculation protocol – Justify the choice of using 100 spores per plant. Indicate whether this inoculum level was determined from preliminary assays, drawn from published literature, or based on standard practice in similar studies.

Response: Our research group has previously conducted experiments on Carica papaya and Tagetes erecta (Figure 1) seedlings, where we observed, based on unpublished data, that inoculating with an initial dose of 100 spores per seedling ensured colonization within the first 15 days. Based on these findings, we chose to apply the same inoculation dose of 100 spores per Annona muricata seedlings.

Figure 1. Behavior of the variables for total colonization percentage (a), Arbuscules (b), Vesicles (c), and Spores (d) in Tagetes erecta plants inoculated with 100 spores of Funneliformis mosseae over time under greenhouse conditions. Unpublished data.

  1. (Q2) Extract concentrations – Clarify the rationale for applying concentrations of 400–800 mg/mL and 0.25 mg/g diet. Specify whether these values were selected according to preliminary dose–response experiments, data from literature, or practical considerations.

Response: We thank the reviewer for the comment. The rationale for extract concentrations has been clarified in the Materials and Methods section. The concentrations of 400–800 mg·mL⁻¹ for T. pallidipennis and 0.25 mg·g⁻¹ diet for S. frugiperda were selected based on first preliminary dose-response analysis, extract yield and practical feasibility for each bioassay, ensuring sufficient sample availability and uniform application across treatments.

  1. (Q3) AMF colonization – Provide information on how the establishment of arbuscular mycorrhizal fungi (AMF) was confirmed. State whether root colonization was evaluated microscopically (e.g., percentage of colonized root length, staining method), as this validation is essential to confirm effective AMF symbiosis.

Response: We thank the reviewer for this valuable comment. The Materials and Methods section was updated to include details on how AMF colonization was verified. Root samples were stained according to Phillips and Hayman (1970) with minor modifications, and the colonization percentage was determined following McGonigle et al. (1990). The presence of hyphae, vesicles, and arbuscules confirmed successful AMF establishment in inoculated plants, while low colonization was detected in the control treatment. Additionally, the text now clarifies that colonization data correspond to those previously reported in González-López et al. (2025a [23]), obtained under identical experimental conditions. These data, along with annonacin concentrations determined in the same study, are included in Table 1 of the present manuscript.

  1. (Q4) Statistical analysis and ethics – Indicate whether the assumptions of parametric statistical tests (normality, homogeneity of variances) were verified and whether non-parametric alternatives were used when assumptions were not satisfied. In addition, include the details of ethical approval for the use of rabbits in Triatoma pallidipennis feeding experiments, specifying the approving committee, protocol number, and compliance with animal welfare standards. Ethical approval for the use of NZW rabbits in the T. pallidipennis feeding experiments was granted by the Ethics Committee for Research (Comité de Ética en Investigación), Coordination of Research and Postgraduate Studies, Centro Universitario del Sur, University of Guadalajara (approval no. CIP/CEI/2024/20). All experimental procedures were performed in accordance with institutional animal welfare guidelines and ethical standards established by the University of Guadalajara. This information has been added to the Materials and Methods section.

Response: We thank the reviewer for this comment. The Statistical analysis section was revised to clarify that data normality and variance homogeneity were verified prior to applying parametric tests. Normality was assessed using the Shapiro–Wilk test and homogeneity of variances with Bartlett’s test. When these assumptions were not satisfied, non-parametric alternatives were applied accordingly.

In the section Results, the following points should be addressed:

  1. (Q1) Tables – While the tables are clear, improvements are needed. In Table 1 (mortality/eggs/annonacin), standard errors or 95% confidence intervals should be added to mortality values. In Table 2 (larval data), mortality is currently presented as percentages only; including the raw number of individuals per treatment would enhance transparency and reproducibility.

Response: We appreciate the reviewer’s valuable suggestion. Standard errors of mortality, calculated from a binomial proportion, have been added to both Table 1 and Table 2. The table footnotes now specify that mortality values are expressed as percentage ± SE (binomial), and the total number of individuals per treatment (n = 25) has been indicated to improve transparency and reproducibility. In addition, standard errors (SEM or SE) have been included for all data presented in both tables.

  1. (Q2) Figures – For Figures 1 and 2, survival curves should be presented with error shading or confidence interval bands, rather than only step curves, to provide a better representation of variability and statistical reliability.

Response: We thank the reviewer for this valuable suggestion. In the revised version, Figures 1 and 2 have been updated to include 95% confidence interval (CI) bands represented as shaded areas around the Kaplan–Meier survival curves. This graphical adjustment provides a clearer visualization of variability and enhances the statistical interpretation of the survival data. The corresponding figure legends were also updated to indicate the inclusion of confidence intervals.

  1. (Q3) Results text – The narrative description of results should not only state that treatments were "effective," but explicitly highlight which treatments differed significantly, supported by statistical comparisons.

Response: We appreciate the reviewer’s suggestion. The Results section has been thoroughly revised to explicitly indicate the statistical differences among treatments and the tests used to support them. Specifically, we now report significant differences in mortality among T. pallidipennis treatments (χ² test, P < 0.05; Table 1), in larval and pupal parameters of S. frugiperda (Kruskal–Wallis, Dunn’s, and Tukey’s tests; P ≤ 0.05; Tables 2 and 3), and in annonacin content among treatments (LSD test; P ≤ 0.05; Section 2.3). Additionally, survival analyses are now supported by Log-Rank (Mantel–Cox) statistics with confidence interval bands (Figures 1 and 2).

  1. (Q4) Annonacin quantification – Although annonacin quantification appears in Table 1, it is not clearly emphasized in the text. A dedicated subsection should be created in the Results to summarize the HPLC quantification of annonacin and to link these findings with the observed bioassay outcomes.

Response: We appreciate the reviewer’s insightful comment. A new subsection entitled “2.3. Annonacin quantification by HPLC” has been added to the Results section. This subsection summarizes the annonacin concentrations determined by HPLC (previously reported in González-López et al., 2025a [23]) and establishes a clear connection between annonacin concentration and insect mortality.

  1. (Q5) Minor typos / taxonomic clarification – Double-check the species name in Table 1. The text currently refers to Meccus pallidipennis, but the correct and accepted name is Triatoma pallidipennis.

Response: We appreciate the reviewer’s observation. The species name has been corrected from Meccus pallidipennis to the currently accepted name Triatoma pallidipennis throughout the manuscript, including Table 1 and related text, to ensure taxonomic accuracy and consistency.

In the section Discussion, the following clarifications are required:

  1. (Q1) Larval biomass paradox – The observed paradoxical increase in larval biomass in some treatments should be clarified. At present, the explanation remains speculative; a more detailed interpretation or possible mechanistic hypotheses are needed.

Response: We thank the reviewer for this valuable observation. The section discussing larval biomass has been expanded to provide a mechanistic interpretation. We now explain that the observed increase in larval biomass under the FM treatment likely reflects a hormetic or compensatory feeding response to dietary stress rather than an improvement in larval health. Sublethal concentrations of insecticidal compounds can induce hormetic effects that stimulate feeding or growth as larvae attempt to offset toxicity (Calabrese & Baldwin, 2002). Additionally, we incorporated findings from Liu et al. (2025), who reported a similar increase in Spodoptera exigua larval size when fed on Perilla frutescens plants inoculated with Funneliformis mosseae, supporting the hypothesis that AMF-mediated metabolic changes may influence insect growth.

  1. (Q2) Potential applications – Expand the discussion on the practical implications of combining AMF and Annona extracts. Specifically, evaluate whether such approaches could replace or complement synthetic insecticides, and acknowledge limitations such as the potential toxicity of annonacins, challenges in extraction scalability, and the need for field-level validation.

Response: We appreciate this valuable comment. The Discussion section was expanded to address the potential applications and practical implications of combining AMF inoculation with Annona-based extracts. The new paragraph emphasizes the possible role of this biotechnological approach as a complementary tool in integrated pest management, while also acknowledging the main limitations related to annonacin toxicity, challenges in large-scale extraction and standardization, and the need for field validation.

  1. (Q3) Comparative context – Include a concise comparison with other plant–AMF systems that are reported to enhance insecticidal secondary metabolites. This would help situate the findings within a broader ecological and applied framework.

Response: We appreciate this insightful suggestion. A new paragraph was added to the Discussion section to provide a comparative context with other AMF–plant systems that enhance secondary metabolism and insecticidal activity. Specifically, examples were included from Mimosa tenuiflora (inoculated with Gigaspora albida), Perilla frutescens (colonized by Rhizophagus irregularis), and Trifolium pratense (inoculated with Claroideoglomus claroideum or mixed AMF consortia). These cases demonstrate that AMF symbiosis can enhance phenolic, volatile, or monoterpene compounds associated with insecticidal or defensive functions, thus situating our findings for A. muricata within a broader ecological and applied framework.

In the section References, the following clarification is required:

  1. (Q1) References formatting – Ensure consistency in the formatting of references. In particular, all species names should be italicized throughout the reference list, following standard taxonomic conventions. Verify that journal names, volume/issue numbers, and page ranges are also presented uniformly.

Response: We appreciate this suggestion. The reference part was homogenized according to the format of the journal.

Reviewer 4 Report

Comments and Suggestions for Authors

While the subject of the present study is interesting, there are methodological shortcomings that question the veracity of the results and the conclusions drawn.  For example, I do not see any data for the plant extracts w/o the AMF that would confirm that the AMF enhanced efficacy of the extracts from infected plants.  More importantly, mortality in the Triatoma bioassay is based on n=20 insects per treatment with no replication.  I would think that at least three biological replicates would be needed to determine significant treatmen differences.  Also, there is hardly enough data from this experiment to state whether annonacin content bears any relationship to bioactivity.  In Table 1, SM has the lowest concentration of annonacin, but is not the least toxic extract.

As for Spodoptera, the only real conclusion here is that FM is more bioactive than the other plant extracts, though less so than cypermethrin.  I note that the text refers to Palgus, but I assume that this is the trade name for cypermethrin.  The Materials and Methods section do not indicate where the material was sourced nor its concentraton of a.i.  While the study as presented may be suitable as a contribution to a Masters thesis, I don't believe it is up to a minimum standard for an international journal.

Author Response

Reviewer 4

While the subject of the present study is interesting, there are methodological shortcomings that question the veracity of the results and the conclusions drawn.  For example, I do not see any data for the plant extracts w/o the AMF that would confirm that the AMF enhanced efficacy of the extracts from infected plants.  More importantly, mortality in the Triatoma bioassay is based on n=20 insects per treatment with no replication.  I would think that at least three biological replicates would be needed to determine significant treatmen differences.  Also, there is hardly enough data from this experiment to state whether annonacin content bears any relationship to bioactivity.  In Table 1, SM has the lowest concentration of annonacin, but is not the least toxic extract.

As for Spodoptera, the only real conclusion here is that FM is more bioactive than the other plant extracts, though less so than cypermethrin.  I note that the text refers to Palgus, but I assume that this is the trade name for cypermethrin.  The Materials and Methods section do not indicate where the material was sourced nor its concentraton of a.i.  While the study as presented may be suitable as a contribution to a Masters thesis, I don't believe it is up to a minimum standard for an international journal.

Response: We thank the reviewer for these important observations. The experimental design included a non-mycorrhizal control (SM) to allow direct comparison between mycorrhizal and non-mycorrhizal plants. Data for this treatment are presented in Table 1 and Table 2. Statistical analyses confirmed that extracts from AMF-inoculated plants had higher annonacin content and (particularly AD, RI, and FM) caused higher mortality than the non-mycorrhizal control. To improve clarity, we revised the Results section to explicitly highlight these comparisons.

We thank the reviewer for this valuable observation. We acknowledge that additional biological replicates would further strengthen statistical inference. However, the bioassays were conducted using the full number of adult T. pallidipennis individuals available at the time of experimentation, as colony maintenance and ethical restrictions on insect handling limited the number of specimens. Each insect was treated and monitored individually and thus considered an independent experimental unit.
Non-parametric statistical analyses (Chi-square, Log-Rank, and Kruskal–Wallis tests) were used to account for the unreplicated design and ensure robust detection of treatment effects

We agree with the reviewer that annonacin alone may not fully explain the insecticidal activity observed. Although treatments with higher annonacin concentrations (RI and AD) tended to show greater mortality in T. pallidipennis, the lack of a strictly linear relationship suggests that additional metabolites could also contribute to bioactivity. We have revised the Discussion to clarify that the observed effects likely reflect AMF-mediated shifts in the overall secondary metabolite profile of A. muricata, rather than changes in annonacin content alone.

We apologize for the confusion. Palgus® is not cypermethrin, but a commercial formulation containing spinetoram (60 g a.i. L⁻¹), used here as a reference bioinsecticide for comparison with plant extracts. In contrast, cypermethrin was applied separately using the product Cipermetrina 200 CE TM (200 g a.i. L⁻¹), which served as the positive control in the T. pallidipennis bioassay. Both formulations were obtained from local agricultural suppliers in Mexico. This clarification has been added to the Materials and Methods section to specify the commercial source and concentration of active ingredient (a.i.) for both reference insecticides.

Round 2

Reviewer 1 Report

Comments and Suggestions for Authors

The revised version of the manuscript shows substantial improvement compared to the initial submission. The authors have carefully addressed the previous comments and clarified the experimental design, data interpretation, and presentation of results. The figures and tables are now clear and appropriately labeled, and the discussion effectively connects the findings to relevant literature. Minor language polishing could further enhance readability, but overall, the manuscript is scientifically sound and clearly written. 

Author Response

Comment: The revised version of the manuscript shows substantial improvement compared to the initial submission. The authors have carefully addressed the previous comments and clarified the experimental design, data interpretation, and presentation of results. The figures and tables are now clear and appropriately labeled, and the discussion effectively connects the findings to relevant literature. Minor language polishing could further enhance readability, but overall, the manuscript is scientifically sound and clearly written.

Response: Following your suggestion, we have carefully reviewed the text once more and performed additional minor language polishing to further enhance clarity and readability.

Reviewer 2 Report

Comments and Suggestions for Authors
  1. Line 79-81: "These species were chosen because they address complementary contexts, thereby linking crop protection and human health within a unified biological framework..":

Delete it

Author Response

Line 79-81: "These species were chosen because they address complementary contexts, thereby linking crop protection and human health within a unified biological framework.":

Delete it

Response: We thank the reviewer for the suggestion. The indicated sentence has been deleted from the manuscript to improve focus and conciseness in the Introduction section.

Reviewer 3 Report

Comments and Suggestions for Authors

I recommend minor revisions to improve the clarity and consistency of the manuscript. Specifically, please consider adding higher-quality and clearer versions of Figures 1 and 2. Additionally, the section concerning ethanol extracts should be clarified, as the study mainly focuses on ethanolic extracts, whereas Figure S1 refers to methanol extracts—this discrepancy could lead to confusion and inconsistency in interpreting the results. Finally, in the Supplementary Materials, ensure that the scientific name A. muricata is written in italics to maintain correct scientific formatting.

Author Response

I recommend minor revisions to improve the clarity and consistency of the manuscript. Specifically, please consider adding higher-quality and clearer versions of Figures 1 and 2. Additionally, the section concerning ethanol extracts should be clarified, as the study mainly focuses on ethanolic extracts, whereas Figure S1 refers to methanol extracts—this discrepancy could lead to confusion and inconsistency in interpreting the results. Finally, in the Supplementary Materials, ensure that the scientific name A. muricata is written in italics to maintain correct scientific formatting.

Response: We appreciate the reviewer’s careful observation. Methanol was used exclusively for annonacin extraction prior to HPLC quantification, due to its higher analytical compatibility with the chromatographic system and its ability to ensure complete solubilization of the standard. In contrast, all bioassays were conducted using ethanolic leaf extracts, as described in Section 4.2, to ensure biosafety and ecological relevance for potential biopesticide applications. This clarification has been added to Section 4.4.2 (Annonacin quantification by HPLC) as follows:

“For quantitative analysis, annonacin was extracted with methanol instead of ethanol because methanol shows better compatibility with the chromatographic system and ensures complete solubilization of the standard. This analytical extraction was used only for HPLC determination, whereas all biological assays were performed with ethanolic leaf extracts described in Section 4.2.”

Additionally, a bridging sentence was added at the end of Section 4.2 (Preparation of soursop leaf extracts) to improve continuity:

“These ethanolic extracts were used in all bioassays; a separate methanolic extraction was performed exclusively for HPLC quantification (see Section 4.4.2).”

Reviewer 4 Report

Comments and Suggestions for Authors

Authors have made a conscientious effort to address the comments/critique of the original manuscript which have resulted in a greatly improved paper.  While the results presented remain open to alternative interpretation, the authors have more robustly supported their conclusions with relevant statistical analyses.

Author Response

Authors have made a conscientious effort to address the comments/critique of the original manuscript which have resulted in a greatly improved paper.  While the results presented remain open to alternative interpretation, the authors have more robustly supported their conclusions with relevant statistical analyses.

Response: We sincerely thank the reviewer for the positive evaluation of our revised manuscript